# Connecting the Dots: What Graph-Based Text Representations Work Best for Text Classification using Graph Neural Networks?

**Margarita Bugueño** and **Gerard de Melo**
Hasso Plattner Institute (HPI) / University of Potsdam
Potsdam, Germany
{margarita.bugueno, gerard.demelo}@hpi.de

## Abstract

Given the success of Graph Neural Networks (GNNs) for structure-aware machine learning, many studies have explored their use for text classification, but mostly in specific domains with limited data characteristics. Moreover, some strategies prior to GNNs relied on graph mining and classical machine learning, making it difficult to assess their effectiveness in modern settings. This work extensively investigates graph representation methods for text classification, identifying practical implications and open challenges. We compare different graph construction schemes using a variety of GNN architectures and setups across five datasets, encompassing short and long documents as well as unbalanced scenarios in diverse domains. Two Transformer-based large language models are also included to complement the study. The results show that i) although the effectiveness of graphs depends on the textual input features and domain, simple graph constructions perform better the longer the documents are, ii) graph representations are especially beneficial for longer documents, outperforming Transformer-based models, iii) graph methods are particularly efficient at solving the task.

## 1 Introduction

Document comprehension involves interpreting words that can alter the meaning of the text based on their placement. For example, in the sentence *"the movie was boring, but I was surprised by the ending"*, the word *but* contrasts ideas. While traditional vector-based text representation methods lack the ability to capture the structural information of a text effectively, graph-based representation strategies explicitly seek to model relationships among different text elements (nodes) through associations between pairs of them (edges), capturing dependencies between text units and leveraging language structure.

While such ideas have a long history (Hassan and Banea, 2006; Mihalcea and Tarau, 2004, *inter*

*alia*), the rise of Graph Neural Network (GNN) models in recent years has made it particularly appealing to convert even unstructured data into graphs. The model can then capture relevant patterns while accounting for dependencies between graph nodes via message passing

For text classification, numerous graph-based text representation schemes have been proposed and demonstrated the efficacy of graphs. However, most of them were designed for particular domain-specific tasks and validated only on short documents using a restricted set of model architectures (Yao et al., 2019; Huang et al., 2022; Wang et al., 2023). Moreover, some of these proposals predate the introduction of GNNs and were validated using graph mining or classical machine learning models, making it challenging to determine the applicability and effectiveness of graphs in broader settings (Castillo et al., 2015, 2017).

Text classification increasingly extends beyond simple topic classification tasks, encompassing real-world challenges such as noisy texts, imbalanced labels, and much longer documents consisting of more than just a few paragraphs. Hence, a comprehensive assessment of the merits and drawbacks of different graph representations and methods in more diverse scenarios is needed.

This work presents a thorough empirical investigation of previously proposed graph-based text representation methods, evaluating how graphs generalize across diverse text classification tasks. We analyze their effectiveness with several GNN-based architectures and setups across five prominent text classification datasets from a broad range of domains. Unlike previous work (Galke and Scherp, 2022), our study considers diverse datasets with both short and longer documents, as well as unbalanced classification scenarios. Additionally, we evaluate the efficacy vs. efficiency of the proposals, an aspect usually neglected in previous studies.

For each graph method, we conducted extensive

experiments using 3 types of convolutional layers as different message-passing strategies for 12 GNN architecture variants, each using one out of 4 pretrained word embedding techniques as node feature vector initialization. This allows us to shed light on what are the most successful choices of GNN architectures for learning from them.

Our study finds that graph methods are a competitive and particularly efficient choice for solving classification tasks. This is because GNNs can capture both local and global dependencies between structural components. Therefore, they can capture rich semantic relationships and dependencies that are important for the task. Additionally, unlike many sequence models, GNNs can naturally handle variable-length inputs by operating on the graph structure, without any need to map every data sample to a fixed-sized vector or truncate them at a fixed maximum sequence length. While longer documents can be particularly challenging, our study finds that GNN methods hold particular promise for longer documents, an aspect unexplored in prior research. However, the graph's effectiveness depends on the textual input features and domain. Based on our experimental results, we provide a discussion regarding what graph construction and GNN architecture choice is preferable depending on the task to be solved. Surprisingly, although Transformer-based Language Models (LMs) yield outstanding results for the considered tasks, they often have difficulties converging when dealing with short texts.

The study is structured around three research questions, which are discussed in Section 4:

1. How does the choice of GNN architecture and setup affect the classification effectiveness?
2. What graph construction method is most effective for text classification?
3. Can graphs compete with state-of-the-art sequence classification models?

## 2   Prior Work on Graphs in NLP

Previous graph-based text representation methods can be categorized into three categories based on the nature of the underlying graph structure. Early graph constructions primarily relied on co-occurrence and textual statistical patterns. Subsequently, more advanced representations integrated linguistic features as graph components. Recently, specialized graph constructions have evolved, entailing intricate structures that encompass the uti-

lization of graph neural networks as essential components within the learning framework.

### 2.1   Early Graph Constructions

For graph-based text representation, a simple approach is to consider word co-occurrence within a fixed-size sliding window: Words are modeled as nodes, and two nodes are connected if the respective words co-occur within a window of at most $N$ words. Mihalcea and Tarau (2004) used such co-occurrence graphs for $N \in \{2, \ldots, 10\}$ as a ranking model for keyword extraction. They found smaller $N$ to be preferable, as the connection between words further apart is often weaker. Hassan and Banea (2006) used the same approach with $N = 2$ along with TextRank to replace term frequency weights, and then conducted text classification with classic machine learning models. In most of their experiments, this scheme outperformed using TF-IDF vectors. Rousseau et al. (2015) also used a fixed-size sliding window graph (calling it *graph-of-words*). They cast text classification as a classification problem by applying graph mining to obtain subgraph features to train a classifier.

*Sequence graphs* are another simple scheme with edges reflecting the original order of words in the text (Castillo et al., 2015). The authors used the number of times the corresponding two words appear consecutively in the text as edge weights.

### 2.2   Linguistic Features as Graphs

Other graph construction methods have been proposed. Mihalcea and Tarau (2004) highlighted that multiple text units and characteristics can be considered as vertices depending on the application at hand. They invoked application-specific criteria to define edges, such as lexical or semantic relations. To this end, they also proposed a similarity-weighted graph for sentence extraction. Every node represents an entire sentence, while edges are defined by measuring their content overlap as the number of shared tokens. Although this scheme can be applied in other tasks (text classification or summarization), it tends to yield fairly densely connected graphs, making it difficult to extract local patterns and discern the content of the text.

Given that traditional work in linguistics and computational linguistics often considers tree and graph-structured formalisms as the principal way of analyzing individual sentences, these may also serve as building blocks for document-level representations (Arora et al., 2009; Joshi and Rosé, 2009,

*inter alia*). For instance, a neural parsing model (Dozat and Manning, 2016; Yuan et al., 2021) can infer word dependencies to obtain syntactic dependency trees. However, the overall graph representation becomes rather sparse, as nodes share edges with only a limited number of other units.

## 2.3 Specialized Graph Constructions

Text Graph Convolutional Network (TextGCN; Yao et al. 2019) was one of the first approaches to include a Graph Convolutional Neural Network (GCN) as a classification method. TextGCN proposes a heterogeneous graph construction using words and documents as nodes. However, this means that new documents cannot be processed without re-training. It employs Point-wise Mutual Information (PMI) similarity as an edge weighting function for word pairs and TF-IDF for word-in-document edges. Other proposals also suggested integrating heterogeneous contextual information such as TensorGCN (Liu et al., 2020), HeteGCN (Ragesh et al., 2021), and HyperGAT (Ding et al., 2020). However, such approaches are fairly resource-intensive.

TextLevelGCN (Huang et al., 2019a) creates one graph per input text. The proposal defines every word as a node, which can be duplicated if a word appears more than once in a text. Edges are defined for word nodes within a sliding window using PMI edge weights. Despite promising results, the experiments were limited to very short documents.

GraphIE (Qian et al., 2019) uses a homogeneous scheme based on co-reference, integrating a GCN with an RNN encoder–decoder architecture for tagging and information extraction tasks. Nodes can be defined as words or entire sentences, connected via co-reference and identical mention edges, to account for non-local and non-sequential ties. A downside of this is that prior domain knowledge is required to establish the edge types.

Some studies have brought back the classic co-occurrence graph construction methods, but using a different message passing function based on Gated Recurrent Units (Li et al., 2015; Cho et al., 2014) for updating node feature vectors (Zhang et al., 2020).

MPAD (Nikolentzos et al., 2020) included an extra master node connected to every other node in the graph. Therefore, the network is densely connected, and the structural information is vague during message passing. Text-MGNN (Gu et al., 2023)

proposes a heterogeneous graph construction, introducing topic nodes to enhance class-aware representation learning. However, it has the same limitations as TextGCN.

Alternatively, two inductive models have reported good results on traditional text classification benchmarks, but the improvement is mostly due to the combination of GNN and BERT models (Huang et al., 2022; Wang et al., 2023). Thus, these strategies are resource-intensive, hard to apply to long documents, and beyond the scope of our study.

Since Zhang et al. (2020) outperform Textlevel-GCN despite using the same graph construction, it is clear that the graph construction method and the way patterns are extracted from it are closely related. Hence, an in-depth study analyzing multiple factors in a controlled setting is necessary.

In terms of broader empirical comparisons, one previous study also conducted a comparative analysis of different approaches for text classification to evaluate the necessity of text-graphs. The authors compared multiple Bag of Words (BoW), sequence, and graph models (Galke and Scherp, 2022), arguing that a multi-layer perceptron enhanced with BoW is a strong baseline for text classification. Nevertheless, the authors limited their analysis to standard data collections with only short texts. In contrast, with the aim to study how graphs perform in more challenging scenarios, our study considers a broader range of domains including much longer documents and unbalanced classification contexts. In addition, we assess the balance between the effectiveness and efficiency of the proposals, a facet typically overlooked in prior research.

## 3 Comparing Graph-Based Text Representations

To study the merits of prominent graph-based text representation strategies, we conducted comprehensive experiments on five well-known text classification datasets. For each task, we compare different graph construction schemes using a variety of GNN models to separate the effect of the graph construction strategy from that of the message-passing technique in the model.

### 3.1 Methods

#### 3.1.1 Graph-Based Text Representation

Among the studied techniques, there are some graph construction methods that follow an intuitive construction process. They are based solely on sim-

ple relationships between pairs of nodes and only consider basic co-occurrence statistics if needed. Thus, they do not require a deep understanding of the semantic structure. In the following, we refer to these sorts of networks as **Intuitive Graphs**. Figure 1 illustrates how they work.

**Window-based:** Following Hassan and Banea (2006), given an input text, if a term has not been previously seen, then a node is added to the graph, and an undirected edge is induced between two nodes if they are two consecutive terms in the text.

**Window-based extended:** As for the above construction, but with a window size of three. With this, each word will ultimately be tied to the two previous terms and the two subsequent ones.

**Sequence-weighted:** This strategy (Castillo et al., 2015) defines a directed graph with nodes for words and edges that represent that the two corresponding lexical units appear together in the text sequence and follow the order in which they appear. Additional edge weights capture the number of times that two words appear together, which is intended to reflect the strength of their relationship.

**Sequence simplified:** Inspired by the above, a simplified version omits the edge weights. Thus, the effect of the edge importance function over the pure graph structure can be studied in isolation.

A more sophisticated graph-based text representation strategy requiring a more elaborate graph construction process is also considered.

**TextLevelGCN:** Every word appearing in a text is treated as a node, and edges are defined between adjacent words in a fixed-size window. Unlike the above Intuitive Graphs, TextLevelGCN (Huang et al., 2019b) considers each word token occurrence as a separate node, i.e., it allows multiple nodes if the corresponding term occurs more than once in the text. Therefore, the specific in-context meaning can be determined by the influence of weighted information from its neighbors. The authors further employed PMI as an edge weighting function for the word associations, as in Yao et al. (2019).

### 3.1.2 Mainstream Text Representations

We further considered several mainstream representation schemes, allowing us to better understand how the graph approaches fare in comparison.

**Bag of Words (BoW):** Given a vocabulary of known words, this strategy uses vectors of term frequencies, discarding any information about the order of words in the text.

**Transformer-based LMs:** We also include BERT (Devlin et al., 2018) and Longformer (Beltagy et al., 2020) Transformers as powerful masked language model-based encoders. While BERT has a maximum input length of 512 tokens, the Longformer extends this limit via a modified attention mechanism that scales linearly with sequence length. The latter trait is desirable when comparing LMs to graphs, which use the complete source text. Please note that Transformer-based LMs are included merely as an informative point of reference for comparison.

### 3.2 Datasets

The literature review reveals that many graph-based text representation methods have been evaluated on different datasets. Most of the time, the proposals were each introduced for a specific task domain and validated on text with very restricted characteristics, such as a limited vocabulary and an average document length of up to 221 words (Hassan and Banea, 2006; Yao et al., 2019). Hence, it is unclear how well these approaches can generalize to other kinds of data in different domains and be applied to longer documents.

We assess the generalizability of graph strategies in text classification, including sentiment analysis, topic classification, and hyperpartisan news detection, across balanced and unbalanced scenarios, including longer documents. We utilize five publicly available datasets (see Table 1), with further details provided in Appendix A.

**App Reviews** (Grano et al., 2017) – English user reviews of Android applications for fine-grained sentiment analysis in an imbalanced setting.

**DBpedia** (Zhang et al., 2015) – A dataset for topic classification consisting of Wikipedia articles based on DBpedia 2014 classes (Lehmann et al., 2015).

**IMDB** (Maas et al., 2011) – Movie reviews from the Internet Movie Database for binary sentiment classification.

**BBC News** (Greene and Cunningham, 2006) – A topic classification dataset[1] consisting of 2,225 English documents from the BBC News website.

**Hyperpartisan News Detection (HND)** (Kiesel et al., 2018) – A collection of 645 news articles[2] labeled according to whether it shows blind or unreasoned allegiance to one party or entity. The dataset exhibits a minor class imbalance.

---

[1] http://derekgreene.com/bbc/
[2] https://zenodo.org/HNDrecord

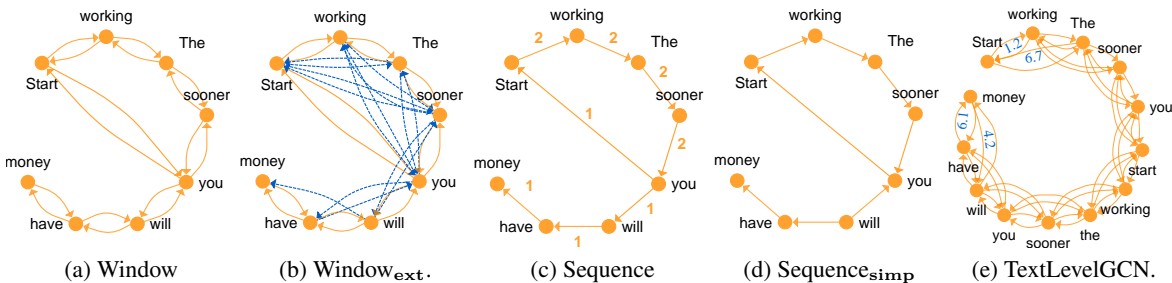

| (a) Window | (b) Window$_{\text{ext}}$. | (c) Sequence | (d) Sequence$_{\text{simp}}$ | (e) TextLevelGCN. |

Figure 1: **Graph Construction Methods.** Given the input text "*Start working! The sooner you start working, the sooner you will have money*", the five co-occurrence graph representations studied are shown. From left to right: window-based graph, window-based graph extended (new edges are shown as dashed in blue), sequence-weighted, sequence simplified omitting edge weights, and TextLevelGCN (edge weights shown for first and last node, in blue).

| Dataset | ADL | K | IR | >512 | >1,024 |
|---|---|---|---|---|---|
| App Reviews | 14 | 5 | 1:8 | 0% | 0% |
| DBpedia | 51 | 14 | 1:1 | 0% | 0% |
| IMDB | 283 | 2 | 1:1 | 12% | 1.4% |
| BBC News | 438 | 5 | 4:5 | 28.5% | 1.6% |
| HND | 912 | 2 | 1:2 | 63.3% | 29.8% |

Table 1: **Statistics of datasets.** This includes the average document length (ADL), the number of classes (K), the imbalance rate between the minority and majority classes (IR), and the proportion of long documents.

## 3.3 Experimental Setup

### 3.3.1 Data Preparation

A fixed-size data partition was taken from each dataset to conduct a fair comparative analysis among the methods. Thus, a training and test split was defined, consisting of 7,000 and 3,000 samples, respectively. For those datasets that did not have that many examples, i.e., BBC News and HND, 80% of the samples were used for training and the remaining 20% for testing. For all datasets, we randomly reserve 10% of the samples from the training set for building the validation set.

Since each graph node represents a word of the input text, a consistent text normalization scheme is needed: We applied lowercase conversion, punctuation mark and stop word removal, as well as eliminating any other non-ASCII characters.

Note that our TextLevelGCN experiments are conducted using the official implementation[3], which incorporates additional preprocessing. This includes removing tokens with fewer than three characters, limiting document lengths to 350 terms, eliminating words with a frequency less than 5, applying lemmatization, as well as applying expansion rules to remove English contractions.

[3]https://github.com/mojave-pku/TextLevelGCN

### 3.3.2 Model Settings

**Graph Neural Networks.** For GNN experiments on Intuitive Graphs, we vary the number of hidden layers from 1 to 4 and vary the dimensionality of node representations in $\{16, 32, 64\}$. We applied Dropout after every convolutional layer with a retention probability of 0.8 and used average pooling for node-level aggregation. The final representation is fed into a softmax classifier.

We compared three types of graph convolutional neural layers: (i) the traditional one (GCN; Kipf and Welling 2016), (ii) using a graph isomorphism operator (GIN; Xu et al. 2018), which has shown improved structural discriminative power compared to GCNs, and (iii) including a graph attentional operator (GAT; Velickovic et al. 2017) with 4 attention heads. Our experiments were based on PyTorch Geometric (see Appendix E).

For TextLevelGCN, we used default parameter settings as in the original implementation, varying the window size ($n$-gram parameter) from 1 to 4.

Four different node vector initialization strategies were also compared. We considered GloVe Wiki-Gigaword 300-dim. embeddings (Pennington et al., 2014), Word2Vec Google News 300-dim. embeddings (Mikolov et al., 2013), static BERT pre-trained embeddings (encoding each token independently and averaging for split terms), and contextualized BERT embeddings. The latter involves encoding the entire input text using BERT and using token embeddings from the 12th layer.

**Bag of Words Baseline.** We employed a cut-off for building the BoW vocabulary by eliminating terms with a document frequency higher than 99% or lower than 0.5%. Once the BoW representations are obtained, a Multi-Layer Perceptron model with

one hidden layer is trained for text classification (**BoW MLP**). We varied the number of hidden units in $\{32, 64, 128, 256\}$ and applied Dropout right before the final classification layer, as done for GNNs.

All GNNs and BoW MLP used a batch size of 64 samples and were trained for a maximum of 100 epochs using Adam optimization (Kingma and Ba, 2014) with an initial learning rate of $10^{-4}$. The training was stopped if the validation macro-averaged F1 score did not improve for ten consecutive epochs. Only for HND, the patience was 20.

**Transformer-based Baselines.** We fully fine-tuned BERT-base uncased, including a Dropout layer right after it with a retention probability of 80%, and a final dense layer for conducting the text classification. During training, the batch size and learning rate were set to 32 and $10^{-6}$, respectively. The maximum number of epochs was 10, and the patience was 5. The same procedure was followed for Longformer-base[4]. However, given the complexity of the model (148 M trainable parameters) and computing resource constraints, the maximum sequence length was set to 1,024 tokens, and the batch size was set to 16.

**General Setup.** The objective function of each model was to minimize the cross-entropy loss. Supplementary experimental details are provided in Appendix A, Appendix C, and Appendix E. For reproducibility, we release our code on `https://github.com/Buguemar/GRTC_GNNs`.

## 4 Results and Analysis

Table 2 and Table 3 show the best architecture and setup for each dataset employing Intuitive Graphs and TextLevelGCN, respectively. The results correspond to the average obtained from 10 independent runs. As some datasets exhibit class imbalance, each table reports the accuracy and the macro-averaged F1-score. The best results are reported in bold, while a star mark is used to indicate the best architecture across the entire dataset. For a full comparison, see Appendix B and Appendix C.

A comparison with baselines such as BERT is given in Table 4.

### 4.1 How do GNN Architecture and Setup Affect the Classification Effectiveness?

**GNN Message Passing.** Table 2 shows GAT as the most effective strategy for DBpedia, IMDB,

---

[4]`https://github.com/allenai/longformer`

and BBC News, compared to other convolutional layers. Due to its attention mechanism, GAT can identify those nodes that are relevant for the final prediction. GAT models also proved to be more robust to variations in parameters such as the number of layers and the hidden units (Appendix B).

However, for imbalanced classification with very short texts (as on App Reviews), GAT is not as effective. In such settings, the graphs have very few nodes, and the attention heads appear to fail to identify the most pertinent ones. Similarly, GAT struggled on HND: Although HND contains extremely long documents and thus there are sufficient elements to exploit, many of the tokens are HTML and PHP markers, or similar source artifacts. Thus, much of the input is insignificant for the task and the attention heads fail to identify relevant nodes. GIN proves to be the best choice for such cases, exploiting the graph structural information for superior discriminative power over traditional GCNs (Xu et al., 2018). While GCNs use simple averages of neighboring node representations, GIN defines a weighted average by learning to determine the importance of a node compared to its neighboring nodes ($\epsilon$-value), which is then fed into an MLP. Thus, GIN can distinguish node neighborhoods, discerning structural information among graph classes. Since our document graphs are based on word co-occurrence, GIN can exploit structural regularities and identify recurrent associations between specific words, which can be crucial for predicting the correct graph-level label.

**Node Feature Initialization.** A noteworthy finding is that the best results were mostly obtained with non-BERT initializations. Well-known static word embeddings with a much lower dimensionality appear to yield better results than BERT embeddings. This is the case for App Reviews and IMDB using Word2Vec, and BBC News using GloVe. Similarly, when using TextLevelGCN as an elaborated graph construction, Word2Vec obtained better results than BERT initialization for some tasks. Moreover, a 1-gram graph construction is sufficient for medium and long text classification when using such an initialization strategy. However, denser graphs are required for short texts.

**Convolutional layers.** The results indicate that the optimal number of convolutional layers is task-dependent, with 1 or 2 layers favored for tasks centered on local patterns and more layers nec-

| Dataset | Emb. | L-Conv | #U | Window | | Window$_{ext}$ | | Sequence | | Sequence$_{simp}$ | |
|---|---|---|---|---|---|---|---|---|---|---|---|
| | | | | Acc | F1-ma | Acc | F1-ma | Acc | F1-ma | Acc | F1-ma |
| App Reviews | Word2Vec | 3-GIN | 16 | **64.7** | 31.0 | **63.6** | 33.9 | **63.3** | 26.4 | 65.3 | 29.1 |
| | | | 32 | 62.0 | 34.9 | 63.2 | 35.0 | 62.0 | 31.0 | ⋆63.7 | ⋆35.7 |
| | | | 64 | 61.1 | **35.1** | 62.4 | **35.4** | 60.0 | **33.0** | 62.5 | 34.8 |
| DBpedia | BERT-C | 1-GAT | 16 | ⋆**97.5** | ⋆**97.4** | 97.3 | 97.3 | 97.3 | 97.2 | 97.3 | 97.3 |
| | | | 32 | 97.2 | 97.2 | **97.3** | 97.2 | 97.0 | 96.9 | 97.0 | 97.0 |
| | | | 64 | 97.1 | 97.1 | 97.1 | 97.1 | 96.7 | 96.7 | 97.0 | 96.9 |
| IMDB | Word2Vec | 1-GAT | 16 | 87.3 | 87.3 | **87.3** | **87.3** | 87.7 | 87.7 | ⋆**87.9** | ⋆**87.9** |
| | | | 32 | 87.3 | 87.3 | 86.9 | 86.9 | 87.5 | 87.5 | 87.5 | 87.5 |
| | | | 64 | **87.4** | 87.3 | 86.7 | 86.6 | 87.2 | 87.2 | 87.4 | 87.4 |
| BBC News | GloVe | 4-GAT | 16 | **97.8** | **97.7** | ⋆**98.0** | ⋆**98.0** | 97.8 | 97.8 | 97.4 | 97.3 |
| | | | 32 | **97.8** | **97.7** | 97.6 | 97.6 | 97.8 | 97.7 | **97.4** | **97.3** |
| | | | 64 | **97.8** | **97.7** | ⋆**98.0** | ⋆**98.0** | 97.6 | 97.5 | 97.2 | 97.1 |
| HND | BERT | 2-GIN | 16 | **77.6** | **76.8** | 75.2 | 73.9 | 56.6 | 36.1 | 77.4 | 76.5 |
| | | | 32 | 75.3 | 73.6 | **77.4** | **76.8** | 56.6 | 36.1 | 78.3 | 77.6 |
| | | | 64 | 77.1 | 76.5 | 76.9 | 75.8 | 56.6 | 36.1 | ⋆**79.1** | ⋆**78.5** |

Table 2: **Best-performing GNN for Intuitive Graphs.** The node feature initialization (Emb.) and architecture details are reported. L-Conv and #U stand for the hidden convolutional layer and units, respectively. The results report the average obtained from 10 independent runs. Full comparison in Appendix B.

| Dataset | Emb. | 1-gram | | 2-gram | | 3-gram | | 4-gram | |
|---|---|---|---|---|---|---|---|---|---|
| | | Acc | F1-ma | Acc | F1-ma | Acc | F1-ma | Acc | F1-ma |
| App Reviews | Word2Vec | **66.6** | 34.7 | 64.7 | 35.2 | ⋆64.5 | ⋆**35.8** | 64.3 | 35.5 |
| DBpedia | BERT | 95.7 | 95.7 | ⋆**96.1** | ⋆**96.0** | 95.9 | 95.9 | 96.0 | **96.0** |
| IMDB | Word2Vec | ⋆**86.8** | ⋆**86.8** | 86.5 | 86.4 | 86.2 | 86.2 | 86.1 | 86.1 |
| BBC News | BERT | 97.0 | 97.0 | 97.2 | 97.2 | ⋆**97.3** | ⋆**97.3** | 97.0 | 97.0 |
| HND | Word2Vec | ⋆**75.7** | ⋆**73.4** | 71.6 | 67.9 | 72.2 | 69.8 | 70.4 | 67.0 |

Table 3: **Best-performing TextLevelGCN.** Results for the best node feature initialization (Emb.). The results report the average obtained from 10 independent runs. Full comparison in Appendix B.

essary for tasks requiring broader information. The contextual understanding, whether local or global, is also influenced by the document length. For instance, to comprehensively grasp the document's sentiment, a sentence-level analysis is vital, whereas if the document comprises only one or two sentences, a wider document-level view is preferable. This is shown in Table 2 and Table 5, where using 3 layers produced the best App Reviews results.

## 4.2 What Graph Construction Method is Most Effective for Text Classification?

**Intuitive Graphs.** The sequence construction in general shows worse performance than its simplified version, which indicates that the use of discrete weights in the edges does not provide relevant information for datasets such as App Reviews, DBpedia, and IMDB. BBC News appears to be an exception: Since news articles tend to reiterate key facts in the news multiple times, exact co-occurrences of word pairs appear to be frequent and might be meaningful. Despite also consisting of news articles, HND behaves similarly to other datasets in that Sequence$_{simp}$ significantly outperforms the weighted version, which fails to learn the task. This may be due to noisy tokens such as HTML tags that may occur numerous times. When omitting edge weights, the model may be less affected by such noise.

Regarding the window-based graph construction, the extended version does not show a significant improvement over the base version with $N = 2$. This is because a higher $N$ increases the average degree of the graph, making it difficult to extract local patterns and discern the content of the text. Hence, Window mostly outperformed Window$_{ext}$.

Overall, the window-based construction is recommended when the classification task is as simple as topic recognition. This allows a faster and more direct identification of the input document's vocabulary, as each token accesses both its left and right context immediately and can identify recurrent words. Moreover, a quick vocabulary exploration is achieved as $N$ grows.

In contrast, for tasks such as sentiment analysis or identifying writing styles and biases in a given article, a detailed analysis of the term order is necessary. In this case, a sequence-based construction seems preferable. Although directed graphs may be

| Dataset | Model | Node Init. | Acc | F1-ma | Exec. Time $[s]$ | #Params |
|---------|-------|------------|-----|-------|------------------|---------|
| App Reviews | BoW MLP | - | **64.7 ± 0.3** | 32.7 ± 0.7 | 104.4 | 10.3 K |
| | BERT | - | 62.0 ± 1.2 | † 36.9 ± 1.1 | 1,891.8 | 108 M |
| | Longformer | - | 63.5 ± 0.9 | **37.6 ± 0.8** | 5,552.2 | 148 M |
| | TextLevelGCN | Word2Vec | † 64.5 ± 1.2 | 35.8 ± 1.0 | 546.4 | 561 K |
| | Sequence$_{simp}$ | Word2Vec | 63.7 ± 0.7 | 35.7 ± 1.3 | 168.8 | 16.3 K |
| DBpedia | BoW MLP | - | 91.5 ± 0.2 | 91.5 ± 0.2 | 24.5 | 52.4 K |
| | BERT | - | **98.3 ± 0.1** | **98.3 ± 0.1** | 2,201.2 | 108 M |
| | Longformer | - | † 98.1 ± 0.2 | † 98.1 ± 0.2 | 5,451.9 | 148 M |
| | TextLevelGCN | BERT | 96.1 ± 0.1 | 96.0 ± 0.2 | 426.8 | 4.8 M |
| | Window | BERT-C | 97.5 ± 0.1 | 97.4 ± 0.1 | 384.3 | 50.3 K |
| IMDB | BoW MLP | - | 83.7 ± 0.2 | 83.7 ± 0.2 | 40.8 | 192 K |
| | BERT | - | † 88.4 ± 0.7 | † 88.4 ± 0.8 | 1,640.1 | 108 M |
| | Longformer | - | **90.5 ± 0.6** | **90.5 ± 0.6** | 4,645.4 | 148 M |
| | TextLevelGCN | Word2Vec | 86.8 ± 0.2 | 86.8 ± 0.3 | 1,022.3 | 10.9 M |
| | Sequence$_{simp}$ | Word2Vec | 87.9 ± 0.1 | 87.9 ± 0.1 | 473.6 | 19.5 K |
| BBC News | BoW MLP | - | 97.9 ± 0.1 | 97.8 ± 0.1 | 8.4 | 329 K |
| | BERT | - | 97.8 ± 0.3 | 97.7 ± 0.3 | 398.9 | 108 M |
| | Longformer | - | **98.2 ± 0.3** | **98.2 ± 0.3** | 1,470.5 | 148 M |
| | TextLevelGCN | BERT | 97.3 ± 0.4 | 97.3 ± 0.4 | 684.2 | 9.6 M |
| | Window$_{ext}$ | GloVe | † 98.0 ± 0.3 | † 98.0 ± 0.3 | 170.6 | 32.6 K |
| HND | BoW MLP | - | 75.6 ± 1.2 | 74.5 ± 1.4 | 5.4 | 444 K |
| | BERT | - | 72.6 ± 2.9 | 70.6 ± 4.5 | 346.1 | 108 M |
| | Longformer | - | † 77.2 ± 3.8 | † 75.5 ± 6.1 | 475.1 | 148 M |
| | TextLevelGCN | Word2Vec | 75.7 ± 2.6 | 73.4 ± 3.5 | 426.8 | 3.2 M |
| | Sequence$_{simp}$ | BERT | **79.1 ± 1.1** | **78.5 ± 1.1** | 116.3 | 66.1 K |

Table 4: **General performance.** The average results over 10 runs for graph models and sequential baselines are reported (see Appendix C). For each model, the average total execution time as well as the number of trainable parameters (#Params) are specified. The best result is shown in bold, while the second best is marked with a † symbol. Please note that the graph construction step is included in the execution time calculation (see Appendix D).

limited to a left-to-right construction, GNNs spread the node information between neighbors and thus exploit structural and linguistic textual features, as local and global contexts of the document.

**TextLevelGCN.** Table 3 shows that TextLevel-GCN is the best-performing graph-based model for App Reviews, implying that the task benefits from edge weights, but that they should be soft values for a smoother learning curve. Otherwise, it is preferable to omit them by employing a Sequence$_{simp}$ construction. Nonetheless, TextLevelGCN underperforms Intuitive Graphs on all other datasets, even when processing medium-length documents.

As in Table 2, for TextLevelGCN there is a connection between the classification task and node feature initialization. Topic classification tasks obtained better results when employing BERT for 2-gram and 3-gram setups. Since vocabulary exploration is relevant to solve the task, an extended left–right context graph construction is beneficial. Likewise, since BERT embeddings are high-dimensional vectors, they are more valuable than other strategies. In turn, the best results for sentiment analysis and detection of biased writing were obtained by 1-gram graphs using Word2Vec. In these cases, only 300 dimensions are sufficient to

get competitive results. Given that App Reviews documents are extremely short, the local context in the text is insignificant and exploring the global context through denser 3-gram graphs is required.

## 4.3 Can Graphs Compete with State-Of-The-Art Sequence Models?

Although graphs do not attain the results of Transformer-based ones for short and medium-length document classification, Intuitive Graphs perform better the longer the documents are. Graph representations are designed to harness the text's structure, and as such, their performance is expected to excel in longer documents as there is more information and structural patterns to exploit.

For BBC News, Window$_{ext}$ has the second-best accuracy at only 0.2 points behind the best-performing model, Longformer. Intuitive Graphs dominate as the best way to represent longer documents (HND). For this scenario, there is a noticeable gap between the best and the second-best model. Therefore, graph-based document representations appear to provide clear advantages when processing long texts. Note that in this task, TextLevelGCN performs better than BERT but worse than BoW MLP. This suggests that, despite

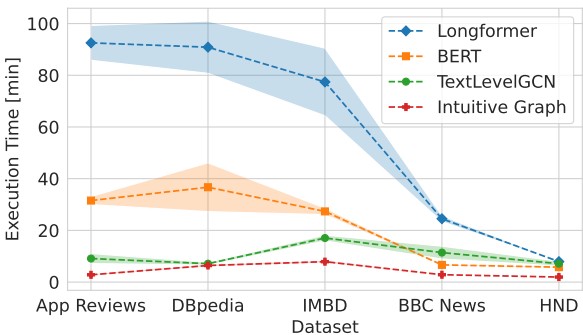

Figure 2: **Execution time.** Average execution time and shaded standard deviation. Time is shown in minutes.

its effectiveness, TextLevelGCN loses a significant part of the input document by defining a much smaller maximum length for text documents (350 tokens). BoW MLP represents each document by considering the entire dataset's vocabulary, granting access to terms beyond TextLevelGCN's scope.

One of the strongest aspects of Intuitive Graphs methods is that they require much less time and compute resources than popular alternatives during training. Although an extra step is required to create document graph representations, the results indicate that the total execution time, including graph creation and model execution, is not an issue. For short texts as in DBpedia, e.g., the window graph is on par with the top-performing LLM, with just a 0.8% accuracy difference and 5.7 times faster speed. Likewise, BERT beats Sequence graphs on IMDB by only 0.5% in accuracy, while being 3.5 times slower. Note that BoW MLP is not included in Figure 2, since it did not obtain good results.

In contrast, since BERT and Longformer are highly complex models in terms of the number of learnable parameters, a higher execution time than for graph-based models is expected. Interestingly, shorter documents, such as those in App Reviews and DBpedia, take even longer than medium-length documents. This suggests that the models require several iterations to converge on these particular tasks. Beyond this, note the abrupt decrease in the execution time for the BBC and HND datasets is because they have a small number of samples. Therefore, the total runtime is much shorter compared to the others. See Appendix D for more details on the runtime and resource utilization.

### 4.4 Discussion

The results show that graph-based document representation holds promise as a way of providing struc-

tural information to deep neural networks. Graph-based learning models are powerful and allow the extraction of complex patterns from text. However, they are particularly task-sensitive and depend on the lexical features of the documents to be represented. Thus, special care must be taken to properly define the components of the structure (nodes, edges, and the similarity function as edge label). Despite this, the most simplistic graph constructions can address text classification fairly well, proving competitive even in challenging scenarios such as with data imbalance and noisy documents.

An interesting finding is that when the focus of the text classification task is on the vocabulary, the global context is much more relevant than the local context of the document. Thus, the best graph construction strategies are those based on extended co-occurrence windows, yielding denser graphs. On the other hand, when the focus is on understanding the document as a whole and how the various parts of the text are connected, the local context becomes much more valuable. Therefore, Window (N=2) or Sequential graphs are recommended.

## 5 Conclusion

We present an empirical analysis of graph representations for text classification by comprehensively analyzing their effectiveness across several GNN architectures and setups. The experiments consider a heterogeneous set of five datasets, encompassing short and long documents. The results show that the strength of graph-based models is closely tied to the textual features and the source domain of documents. Thus, the choice of nodes and edges is found to be crucial. Despite this, Intuitive Graphs are shown to be a strong option, reaching competitive results across all considered tasks, especially for longer documents, exceeding those of BERT and Longformer. Additionally, we observed that pre-trained static word embeddings, instead of BERT vectors, allow reaching outstanding results on some tasks.

We are enthusiastic about extending our study to further tasks in future work. To this end, we are releasing our code on GitHub[5] and hope that it can grow to become a community resource. Additionally, we will expand this study by exploring approaches for learning the graph structure to eliminate the need for picking a design manually, being less domain-dependent.

---

[5] https://github.com/Buguemar/GRTC_GNNs

## Limitations

While this study successfully shows the impact and potential of graphs for document representation, there are some points to keep in mind.

First, despite all the judgments and conclusions presented being supported by the results of the experiments, they were based on graph neural network models trained on particular sub-partitions, as stated in Section 3.3.1, so as to allow a fairer comparison between models. However, this means that the results reported here are not directly comparable with those reported in the literature. To assess how the models are positioned with regard to the state-of-the-art in the different tasks, it is advisable to train on the original training partitions and thus learn from all the available data.

It is also important to note that our study analyzes multiple text representation strategies on text classification only. Although this is one of the most important classes of NLP tasks, we cannot ensure that such graph approaches show the same behavior in other tasks. Therefore, tackling other types of problems that require a deep level of understanding of the local and global context of the text is an important direction for future work.

Finally, all the experiments were run on English data. As English has comparatively simple grammar and well-known rules for conjugations and plurals, it is possible that graph-based models may not be as effective in other languages. Analyzing this aspect would be particularly interesting for low-resource languages.

## Ethics Statement

This work studies fundamental questions that can be invoked in a multitude of different application contexts. Different applications entail different ethical considerations that need to be accounted for before deploying graph-based representations. For instance, applying a trained hyperpartisan news detection model in an automated manner bears the risk of false positives, where legitimate articles get flagged merely for a choice of words that happens to share some resemblance with words occurring in hyperpartisan posts. For sentiment classification, Mohammad (2022) provides an extensive discussion of important concerns. Hence, ethical risks need to be considered depending on the relevant target use case.

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

## A  Dataset Descriptions

We provide a detailed description of the datasets used for our text classification experiments. All of them were labeled by experts and validated by the community.

**App Reviews.** The dataset is a collection of 288,065 English user reviews of Android applications from 23 different app categories (Grano et al., 2017). The goal of the dataset is the fine-grained sentiment analysis in an imbalanced setting, where 60.5% of the total samples correspond to 4-star reviews. Each example includes the name of the software application package, the comment, the date when the user posted the evaluation, and the rating provided.

**DBpedia.** For topic classification, the DBpedia ontology classification dataset (Zhang et al., 2015) was constructed by picking 14 non-overlapping classes from DBpedia 2014 (Lehmann et al., 2015). For each category, the authors randomly chose 40,000 Wikipedia articles as training samples and 5,000 samples for testing. Every article contains the title, content, and class label. Although the original DBpedia is a multilingual knowledge base, this dataset only contains English data.

**IMDB.** English language movie reviews from the Internet Movie Database for binary sentiment classification (Maas et al., 2011). The dataset is composed of 25,000 reviews for training and 25,000 for testing, with balanced numbers of positive and negative reviews.

**BBC News.** This is a publicly available[6] dataset consisting of 2,225 English documents from the BBC News website (Greene and Cunningham, 2006). The articles correspond to stories from 2004–2005 in the areas of business, entertainment, politics, sport, and technology. The dataset exhibits minor class imbalance, with sports being the majority class with 511 articles, while entertainment is the smallest one with 386 samples.

**Hyperpartisan News Detection (HND).** A dataset[7] for binary news classification (Kiesel et al., 2018). Although it comprises two parts, *byarticle* and *bypublisher*, this study only uses the first one. The dataset has 645 English samples labeled through crowdsourcing, with 238 (37%) labeled as hyperpartisan and 407 (63%) as not hyperpartisan. The challenge of this task is to detect the hyperpartisan language, which may be distinguishable from regular news at the levels of style, syntax, semantics, and pragmatics (Kiesel et al., 2019).

## B  Word Embeddings for Node Initialization

In the following, we provide further more detailed investigations pertaining to the choice of word embeddings to initialize node representations.

### B.1  Intuitive Graphs

We include the results reported by the GNN models trained on the different datasets using four different node feature initialization strategies.

---

[6]BBC full text: `http://derekgreene.com/bbc/`

[7]`https://zenodo.org/HNDrecord`

The results are shown from Table 5 to Table 9 and include BERT pre-trained word embeddings (BERT), contextualized BERT (BERT-C), GloVe, and Word2Vec.

Each table presents the accuracy and macro averaged F1-score as averages over 10 runs. Note that the underlined embedding strategy is the one that attained the best performance, as shown in Table 2 and Table 3.

## B.2 TextLevelGCN

As discussed in Section 3.1, one of the main contributions of TextLevelGCN is that it allows duplicate nodes when a term occurs more than once in the input text. Therefore, it takes care of polysemy. Hence, using the message-passing function, the model can infer the proper meaning of the token given its local context. Given this peculiarity, we exclude contextualized BERT (BERT-C) as a node feature initialization strategy. Thus, the performance of TextLevelGCN was analyzed using BERT pre-trained word embeddings, GloVe, and Word2Vec. Note that the underlined embedding strategy is the one that attained the best performance, as in Table 3. The results are presented in Table 10 and correspond to the average over 10 independent trials.

## C Transformer-based language models

In order to provide results on a broader spectrum regarding the behavior of Transformer-based LMs, we performed additional experiments using the pretrained BERT and Longformer models. The corresponding results are shown in Table 11.

A pre-trained BERT-base uncased model was included by freezing the encoder architecture and stacking a final dense layer for conducting the corresponding text classification, as done for the fully fine-tuned version described in Section 3.3.2. The same process was followed for the pre-trained Longformer-base. In this case, we conducted experiments setting a maximum sequence length of 512, and 1,024. This was done to have a fair comparison regarding BERT and thus separate the effect that attention has on both approaches.

For training, we used Adam optimizer (Kingma and Ba, 2014) with an initial learning rate of $10^{-4}$, a batch size of 64 samples, 100 epochs as a maximum, and early stopping with patience 10. Only for HND dataset, the patience was 20.

## D Runtime & Resource Utilization

Table 12 presents additional information concerning the execution time for graph models. The average total execution time is broken down into graph representation generation time and GNN running time.

All the experiments conducted in this study were run on an NVIDIA RTX A6000 with 48GB VRAM.

To complement the results reported in Table 4, we measured the GPU utilization (%) and GPU memory usage (%) for each of the models. We also measured these metrics for each graph construction when applied to each of the datasets to find out how the strategies behave when scaling to longer documents. We tracked model performance by using Weights & Biases (W&B)[8] platform. We reran all the models using the same batch size for a fair comparison.

Table 13 suggests: i) The increase in GPU utilization is minimal as the document length increases. Specifically, as the document length increases by one order of magnitude, GPU utilization increases by about 1.5% when employing Intuitive Graphs and 8-10% for TLGCN. ii) The GPU memory allocated for graph strategies is constrained to below 6%, representing a mere fifth of the memory consumed by BERT and less than a tenth of the memory consumed by Longformer. This is a significant consideration when computational resources are restricted.

## E Libraries Used

In order to provide the reader and practitioners with the necessary details to regenerate the reported results, Table 14 presents all the libraries used to perform the experiments.

---

[8]https://docs.wandb.ai/guides/app/features/system-metrics

Table 5:

| Emb. | Layers | Units | Window Acc | Window F1-ma | Window_ext Acc | Window_ext F1-ma | Sequence Acc | Sequence F1-ma | Sequence_simp Acc | Sequence_simp F1-ma |
|---|---|---|---|---|---|---|---|---|---|---|
| BERT | 2 | 16 | **63.4** | 32.4 | **59.8** | **32.4** | **59.9** | 23.4 | **63.2** | 32.4 |
| | | 32 | 61.6 | **34.0** | 58.0 | 31.9 | 57.9 | 26.5 | 60.3 | 33.2 |
| | | 64 | 60.3 | 32.0 | 57.9 | 31.9 | 57.1 | **26.6** | 59.3 | **33.4** |
| | 3 | 16 | **63.9** | 29.6 | **60.2** | 31.0 | **59.6** | 23.4 | **64.0** | 27.5 |
| | | 32 | 62.1 | **34.8** | 59.1 | **32.7** | 59.0 | 22.6 | 61.3 | **33.2** |
| | | 64 | 60.0 | 33.9 | 57.8 | 32.6 | 57.3 | **25.1** | 60.5 | **33.2** |
| BERT-C | 2 | 16 | **62.2** | 30.2 | **62.4** | 29.8 | **60.2** | 26.6 | **62.6** | 31.3 |
| | | 32 | 61.1 | **32.0** | 60.0 | **32.5** | 57.1 | **31.3** | 59.5 | **32.1** |
| | | 64 | 58.5 | 31.8 | 58.7 | 31.1 | 56.7 | 30.3 | 58.7 | 31.5 |
| | 3 | 16 | **62.5** | 29.6 | **62.5** | 28.3 | **60.8** | 24.9 | **63.0** | 26.6 |
| | | 32 | 60.4 | **32.4** | 60.1 | 32.2 | 57.8 | **31.0** | 60.6 | **32.2** |
| | | 64 | 59.7 | 31.7 | 60.8 | **32.5** | 56.7 | **31.0** | 58.9 | 32.1 |
| GloVe | 2 | 16 | **63.2** | 31.4 | **63.4** | 32.1 | **63.4** | 27.3 | **64.5** | 31.0 |
| | | 32 | 61.2 | **34.0** | 60.8 | **33.8** | 59.5 | 33.4 | 63.3 | 33.1 |
| | | 64 | 59.6 | 32.9 | 60.2 | 33.0 | 58.3 | **34.3** | 61.2 | **33.8** |
| | 3 | 16 | **64.5** | 28.8 | **63.8** | 30.7 | **63.1** | 27.0 | **64.9** | 28.6 |
| | | 32 | 61.2 | 32.9 | 61.1 | **34.2** | 61.6 | 32.1 | 62.5 | 32.8 |
| | | 64 | 59.8 | **34.3** | 59.7 | 33.5 | 59.0 | **34.4** | 60.4 | **34.4** |
| Word2Vec | 2 | 16 | **64.0** | 32.7 | **64.4** | 33.8 | **63.1** | 28.2 | **64.8** | 33.7 |
| | | 32 | 62.1 | 34.0 | 63.1 | 34.2 | 60.9 | 31.1 | 62.9 | **34.7** |
| | | 64 | 61.7 | **35.0** | 60.9 | **34.5** | 59.9 | **33.4** | 62.2 | 34.3 |
| | 3 | 16 | **64.7** | 31.0 | **63.6** | 33.9 | **63.3** | 26.4 | **65.3** | 29.1 |
| | | 32 | 62.0 | 34.9 | 63.2 | 35.0 | 62.0 | 31.0 | ★63.7 | ★**35.7** |
| | | 64 | 61.1 | **35.1** | 62.4 | **35.4** | 60.0 | **33.0** | 62.5 | 34.8 |

Table 5: **Word embedding (Emb.) effect on App Reviews.** Accuracy and macro averaged F1-score for Intuitive Graphs using a GIN convolutional neural network.

| Emb. | Layers | Units | Window Acc | Window F1-ma | Window_ext Acc | Window_ext F1-ma | Sequence Acc | Sequence F1-ma | Sequence_simp Acc | Sequence_simp F1-ma |
|---|---|---|---|---|---|---|---|---|---|---|
| BERT | 1 | 16 | **95.9** | 95.8 | 95.8 | 95.8 | 95.9 | 95.8 | 95.8 | 95.7 |
| | | 32 | **95.9** | **95.9** | **95.9** | **95.9** | 95.9 | **95.9** | **96.0** | **95.9** |
| | | 64 | 95.8 | 95.8 | **95.9** | **95.9** | **96.0** | **95.9** | 95.9 | 95.9 |
| | 2 | 16 | **95.6** | **95.5** | 95.5 | 95.4 | 95.6 | 95.5 | 95.6 | 95.5 |
| | | 32 | 95.5 | 95.4 | 95.5 | 95.4 | 95.6 | 95.5 | 95.4 | 95.4 |
| | | 64 | 95.3 | 95.3 | 95.2 | 95.1 | 95.3 | 95.3 | 95.3 | 95.3 |
| BERT-C | 1 | 16 | ★**97.5** | ★**97.4** | 97.3 | **97.3** | 97.3 | 97.2 | 97.3 | 97.3 |
| | | 32 | 97.2 | 97.2 | **97.3** | 97.2 | 97.0 | 96.9 | 97.0 | 97.0 |
| | | 64 | 97.1 | 97.1 | 97.1 | 97.1 | 96.7 | 96.7 | 97.0 | 96.9 |
| | 2 | 16 | **97.4** | **97.3** | 97.3 | **97.3** | 97.3 | **97.3** | 97.3 | **97.3** |
| | | 32 | 97.2 | 97.2 | **97.3** | **97.3** | 97.0 | 97.0 | 97.2 | 97.2 |
| | | 64 | 97.3 | 97.2 | **97.3** | **97.3** | 97.0 | 97.0 | 97.1 | 97.0 |
| GloVe | 1 | 16 | **95.9** | **95.9** | 95.9 | 95.8 | 95.8 | 95.7 | **96.0** | **96.0** |
| | | 32 | **95.9** | **95.9** | 96.1 | 96.0 | 96.0 | 95.9 | 96.0 | 96.0 |
| | | 64 | 95.9 | 95.8 | 96.0 | 95.9 | 95.9 | 95.8 | **96.0** | **96.0** |
| | 2 | 16 | **95.9** | 95.8 | 95.8 | 95.8 | 95.9 | 95.9 | 96.0 | 95.9 |
| | | 32 | **95.9** | 95.8 | 95.7 | 95.6 | 95.9 | 95.9 | 96.1 | 96.0 |
| | | 64 | 95.7 | 95.7 | **95.8** | **95.8** | 95.9 | 95.8 | 95.9 | 95.9 |
| Word2Vec | 1 | 16 | 95.9 | 95.8 | 95.7 | 95.6 | **95.7** | **95.7** | 95.8 | 95.8 |
| | | 32 | **96.0** | **96.0** | 95.8 | 95.7 | 95.7 | 95.7 | 95.8 | 95.8 |
| | | 64 | 95.9 | 95.9 | 95.5 | 95.4 | 95.6 | 95.5 | 95.7 | 95.7 |
| | 2 | 16 | **95.6** | **95.5** | 95.4 | 95.3 | 95.6 | 95.5 | 95.7 | 95.6 |
| | | 32 | 95.4 | 95.4 | 95.4 | 95.3 | 95.5 | 95.4 | 95.3 | 95.2 |
| | | 64 | 95.4 | 95.3 | 95.3 | 95.3 | 95.4 | 95.4 | 95.5 | 95.4 |

Table 6: **Word embedding (Emb.) effect on DBpedia.** Accuracy and macro averaged F1-score for Intuitive Graphs using a GAT convolutional neural network.

| Emb. | Layers | Units | Window | | Window$_{ext}$ | | Sequence | | Sequence$_{simp}$ | |
|---|---|---|---|---|---|---|---|---|---|---|
| | | | Acc | F1-ma | Acc | F1-ma | Acc | F1-ma | Acc | F1-ma |
| BERT | 1 | 16 | 86.8 | 86.8 | **86.3** | **86.3** | **86.6** | **86.6** | **86.4** | **86.4** |
| | | 32 | **86.9** | **86.9** | 86.0 | 86.0 | **86.6** | 86.5 | 86.3 | 86.3 |
| | | 64 | 86.7 | 86.7 | 86.0 | 86.0 | 86.3 | 86.2 | 86.3 | 86.3 |
| | 2 | 16 | **86.9** | **86.9** | **86.7** | **86.7** | **86.8** | 86.7 | **86.5** | **86.4** |
| | | 32 | 86.5 | 86.5 | 86.0 | 85.9 | **86.8** | **86.8** | 86.1 | 86.1 |
| | | 64 | 85.7 | 85.7 | 86.3 | 86.2 | 86.2 | 86.1 | 86.2 | 86.2 |
| BERT-C | 1 | 16 | **85.7** | **85.7** | **85.9** | **85.9** | 84.9 | 84.8 | 85.7 | 85.6 |
| | | 32 | 85.6 | 85.6 | 85.5 | 85.5 | **85.4** | **85.4** | 85.5 | 85.5 |
| | | 64 | 85.2 | 85.1 | 85.3 | 85.3 | 85.3 | 85.2 | **85.9** | **85.9** |
| | 2 | 16 | 84.6 | 84.5 | **85.0** | 84.9 | **85.8** | **85.8** | 85.1 | 85.1 |
| | | 32 | 85.2 | 85.2 | 84.9 | **84.9** | 85.3 | 85.3 | **85.3** | **85.2** |
| | | 64 | **85.3** | **85.3** | 84.6 | 84.5 | 85.6 | 85.6 | 85.0 | 84.9 |
| GloVe | 1 | 16 | **85.9** | **85.9** | 85.7 | 85.7 | 86.1 | 86.1 | 85.5 | 85.5 |
| | | 32 | 85.3 | 85.3 | 85.2 | 85.2 | 85.8 | 85.8 | **85.5** | **85.5** |
| | | 64 | 85.1 | 85.1 | 84.7 | 84.7 | 85.6 | 85.6 | 85.4 | 85.4 |
| | 2 | 16 | **85.1** | **85.1** | 84.6 | 84.5 | 86.1 | 86.1 | 86.0 | 86.0 |
| | | 32 | 84.9 | 84.9 | 83.7 | 83.7 | 85.5 | 85.5 | 85.3 | 85.3 |
| | | 64 | 84.7 | 84.7 | 83.7 | 83.6 | 85.2 | 85.1 | 84.7 | 84.6 |
| Word2Vec | 1 | 16 | 87.3 | 87.3 | **87.3** | **87.3** | **87.7** | **87.7** | ⋆**87.9** | ⋆**87.9** |
| | | 32 | 87.3 | 87.3 | 86.9 | 86.9 | 87.5 | 87.5 | 87.5 | 87.5 |
| | | 64 | **87.4** | 87.3 | 86.7 | 86.6 | 87.2 | 87.2 | 87.4 | 87.4 |
| | 2 | 16 | **87.5** | **87.4** | **87.3** | **87.3** | **87.6** | **87.6** | 87.8 | 87.8 |
| | | 32 | 86.9 | 86.9 | 87.1 | 87.0 | 87.0 | 87.0 | 87.3 | 87.3 |
| | | 64 | 86.7 | 86.7 | 86.1 | 86.1 | 87.2 | 87.2 | 86.6 | 86.6 |

Table 7: **Word embedding (Emb.) effect on IMDB.** Accuracy and macro averaged F1-score for Intuitive Graphs using a GAT convolutional neural network.

| Emb. | Layers | Units | Window | | Window$_{ext}$ | | Sequence | | Sequence$_{simp}$ | |
|---|---|---|---|---|---|---|---|---|---|---|
| | | | Acc | F1-ma | Acc | F1-ma | Acc | F1-ma | Acc | F1-ma |
| BERT | 3 | 16 | **96.9** | **96.7** | **97.1** | **97.1** | **97.0** | **96.9** | 96.7 | 96.5 |
| | | 32 | 96.5 | 96.3 | 96.9 | 96.8 | 96.4 | 96.3 | **96.7** | **96.5** |
| | | 64 | 96.5 | 96.3 | 97.0 | 96.9 | **97.0** | 96.8 | **96.7** | **96.5** |
| | 4 | 16 | **96.5** | **96.4** | **96.7** | 96.6 | **96.7** | **96.5** | **96.9** | **96.7** |
| | | 32 | 96.4 | 96.4 | 96.3 | 96.3 | 96.0 | 95.8 | 96.0 | 95.8 |
| | | 64 | **96.5** | **96.4** | **96.7** | **96.7** | 95.8 | 95.6 | 96.4 | 96.2 |
| BERT-C | 3 | 16 | 96.2 | 96.1 | 96.7 | 96.6 | 96.4 | 96.3 | 96.1 | 96.0 |
| | | 32 | 96.1 | 96.0 | **96.8** | **96.7** | 96.5 | 96.3 | **96.8** | **96.7** |
| | | 64 | **97.0** | **96.9** | 96.0 | 96.0 | **96.7** | **96.5** | 96.0 | 95.8 |
| | 4 | 16 | 96.2 | 96.1 | **96.8** | **96.7** | 96.6 | 96.5 | 96.5 | 96.4 |
| | | 32 | 96.4 | 96.3 | **96.8** | **96.7** | 96.5 | 96.4 | 96.4 | 96.3 |
| | | 64 | **96.6** | **96.5** | 96.7 | 96.6 | **96.6** | **96.5** | 96.2 | 96.1 |
| GloVe | 3 | 16 | 97.6 | 97.5 | **98.0** | **97.9** | 97.9 | 97.8 | 97.3 | 97.2 |
| | | 32 | 97.5 | 97.4 | 97.9 | 97.8 | 97.8 | 97.7 | **97.6** | **97.5** |
| | | 64 | **97.7** | **97.6** | 97.6 | 97.6 | 97.7 | 97.6 | 97.3 | 97.2 |
| | 4 | 16 | **97.8** | **97.7** | ⋆**98.0** | ⋆**98.0** | 97.8 | 97.8 | 97.4 | 97.3 |
| | | 32 | **97.8** | **97.7** | 97.6 | 97.6 | 97.8 | 97.7 | **97.4** | **97.3** |
| | | 64 | **97.8** | **97.7** | ⋆**98.0** | ⋆**98.0** | 97.6 | 97.5 | 97.2 | 97.1 |
| Word2Vec | 3 | 16 | 96.9 | 96.8 | **97.5** | **97.4** | 97.3 | 97.2 | 97.1 | 96.9 |
| | | 32 | 97.1 | 97.0 | 97.1 | 96.9 | 97.1 | 96.9 | 97.5 | 97.3 |
| | | 64 | **97.3** | **97.2** | 96.8 | 96.6 | **97.6** | **97.4** | **97.7** | **97.5** |
| | 4 | 16 | 96.9 | 96.8 | 97.5 | 97.3 | 97.3 | 97.2 | 97.2 | 97.0 |
| | | 32 | **97.1** | **97.0** | 97.5 | 97.3 | **97.6** | **97.4** | **97.4** | **97.3** |
| | | 64 | 96.9 | 96.8 | **97.6** | **97.4** | 97.4 | 97.2 | 97.3 | 97.0 |

Table 8: **Word embedding (Emb.) effect on BBC News.** Accuracy and macro averaged F1-score for Intuitive Graphs using a GAT convolutional neural network.

| Emb. | Layers | Units | Window Acc | Window F1-ma | Window$_{ext}$ Acc | Window$_{ext}$ F1-ma | Sequence Acc | Sequence F1-ma | Sequence$_{simp}$ Acc | Sequence$_{simp}$ F1-ma |
|---|---|---|---|---|---|---|---|---|---|---|
| BERT | 2 | 16 | **77.6** | **76.8** | 75.2 | 73.9 | 56.6 | 36.1 | 77.4 | 76.5 |
| | | 32 | 75.3 | 73.6 | **77.4** | **76.8** | 56.6 | 36.1 | 78.3 | 77.6 |
| | | 64 | 77.1 | 76.5 | 76.9 | 75.8 | 56.6 | 36.1 | ★**79.1** | ★**78.5** |
| | 3 | 16 | 76.7 | 75.8 | 74.9 | 73.9 | 56.6 | 36.1 | 73.5 | 70.9 |
| | | 32 | 75.7 | 73.9 | 75.2 | 73.5 | 56.6 | 36.1 | **77.9** | **77.1** |
| | | 64 | **77.2** | **76.6** | 75.6 | 74.6 | 56.6 | 36.1 | 77.3 | 76.1 |
| BERT-C | 2 | 16 | 73.6 | 73.0 | 71.6 | 70.8 | **72.8** | **72.5** | 66.4 | 65.6 |
| | | 32 | **74.0** | **73.6** | **73.1** | **71.4** | 70.2 | 69.3 | 67.7 | 66.4 |
| | | 64 | **74.0** | 73.2 | 71.8 | 70.6 | 70.5 | 69.1 | **67.8** | **66.7** |
| | 3 | 16 | 72.8 | 72.0 | 70.8 | 69.9 | **72.2** | **71.8** | **68.0** | **66.8** |
| | | 32 | **74.3** | **73.6** | **71.9** | **70.8** | 70.5 | 69.4 | 67.1 | 65.4 |
| | | 64 | 72.7 | 72.0 | 71.5 | 70.1 | 70.0 | 69.6 | 66.8 | 65.4 |
| GloVe | 2 | 16 | 73.5 | 71.9 | 70.9 | 69.8 | 68.4 | 66.4 | 70.7 | 69.3 |
| | | 32 | 73.6 | 72.6 | 72.2 | 71.3 | **70.2** | **69.3** | **73.7** | **73.0** |
| | | 64 | **76.7** | **75.9** | **73.9** | **73.0** | 70.2 | 68.8 | 73.0 | 72.3 |
| | 3 | 16 | 74.3 | 72.9 | 69.1 | 68.0 | 66.9 | 63.1 | 74.3 | 73.5 |
| | | 32 | **74.7** | **73.5** | 72.3 | 71.5 | 69.7 | 67.8 | 74.7 | 73.7 |
| | | 64 | 73.7 | 73.0 | **74.3** | **73.4** | **70.8** | **70.1** | **75.0** | **74.4** |
| Word2Vec | 2 | 16 | 73.3 | 73.2 | 74.0 | 73.4 | 59.1 | 42.7 | 72.3 | 71.7 |
| | | 32 | **75.0** | **74.7** | 73.0 | 72.0 | **71.0** | **69.4** | 72.6 | 72.0 |
| | | 64 | 73.1 | 72.7 | **75.6** | **74.9** | 66.0 | 57.8 | **73.5** | **73.2** |
| | 3 | 16 | 73.3 | 72.7 | 74.2 | 73.7 | 59.8 | 43.1 | 72.5 | 71.4 |
| | | 32 | **74.5** | **73.9** | 74.5 | 74.1 | **68.4** | **62.6** | 73.2 | 72.8 |
| | | 64 | 74.0 | 73.5 | **75.0** | **74.5** | 61.4 | 47.6 | **75.3** | **75.0** |

Table 9: **Word embedding (Emb.) effect on HND.** Accuracy and macro averaged F1-score for Intuitive Graphs using a GIN convolutional neural network.

| Dataset | Emb. | 1-gram Acc | 1-gram F1-ma | 2-gram Acc | 2-gram F1-ma | 3-gram Acc | 3-gram F1-ma | 4-gram Acc | 4-gram F1-ma |
|---|---|---|---|---|---|---|---|---|---|
| App Reviews | BERT | **65.9** | 35.0 | 64.1 | 34.5 | **65.9** | **35.0** | 63.4 | 34.6 |
| | GloVe | 65.7 | **34.4** | **65.9** | 34.2 | 65.5 | 34.1 | 64.2 | 33.9 |
| | Word2Vec | **66.6** | 34.7 | 64.7 | 35.2 | ★**64.5** | ★**35.8** | 64.3 | 35.5 |
| DBpedia | BERT | 95.7 | 95.7 | ★**96.1** | ★**96.0** | 95.9 | 95.9 | 96.0 | **96.0** |
| | GloVe | 95.7 | 95.6 | **95.8** | **95.8** | 95.6 | 95.6 | 95.6 | 95.6 |
| | Word2Vec | 95.7 | 95.6 | 95.7 | **95.7** | 95.7 | **95.7** | **95.8** | **95.7** |
| IMDB | BERT | **86.7** | **86.7** | 85.9 | 85.9 | 85.7 | 85.7 | 85.7 | 85.7 |
| | GloVe | 86.2 | 86.2 | 86.4 | 86.4 | **86.7** | **86.7** | 86.4 | 86.4 |
| | Word2Vec | ★**86.8** | ★**86.8** | 86.5 | 86.4 | 86.2 | 86.2 | 86.1 | 86.1 |
| BBC News | BERT | 97.0 | 97.0 | 97.2 | 97.2 | ★**97.3** | ★**97.3** | 97.0 | 97.0 |
| | GloVe | **96.1** | **96.1** | **96.1** | **96.1** | 95.5 | 95.5 | **96.1** | **96.1** |
| | Word2Vec | 96.4 | 96.4 | 96.6 | 96.5 | **96.8** | **96.8** | 96.5 | 96.5 |
| HND | BERT | 74.8 | 70.8 | 74.5 | 71.5 | 72.6 | 69.6 | 74.1 | **71.6** |
| | GloVe | **73.8** | 71.0 | 69.8 | 67.2 | 72.1 | 70.2 | **73.8** | **72.0** |
| | Word2Vec | ★**75.7** | ★**73.4** | 71.6 | 67.9 | 72.2 | 69.8 | 70.4 | 67.0 |

Table 10: **Word embeddings as TextLevelGCN node initialization.** Accuracy and macro averaged F1-score are reported.

| Dataset | Model | Length | FZ | Acc | F1-ma |
|---|---|---|---|---|---|
| App Reviews | BERT | 512 | ✓ | 61.3 | 18.4 |
| | | | - | 62.0 | 36.9 |
| | Longformer | 512 | ✓ | 60.4 | 15.1 |
| | | | - | **64.0** | 36.8 |
| | | 1024 | ✓ | 60.4 | 15.1 |
| | | | - | 63.5 | **37.6** |
| DBpedia | BERT | 512 | ✓ | 74.6 | 74.1 |
| | | | - | **98.3** | **98.3** |
| | Longformer | 512 | ✓ | 82.5 | 81.0 |
| | | | - | 98.2 | 98.2 |
| | | 1024 | ✓ | 82.0 | 80.3 |
| | | | - | 98.1 | 98.1 |
| IMDB | BERT | 512 | ✓ | 68.6 | 67.7 |
| | | | - | 88.4 | 88.4 |
| | Longformer | 512 | ✓ | 77.9 | 77.7 |
| | | | - | 90.3 | 90.3 |
| | | 1024 | ✓ | 78.2 | 78.0 |
| | | | - | **90.5** | **90.5** |
| BBC News | BERT | 512 | ✓ | 52.5 | 49.3 |
| | | | - | 97.8 | 97.7 |
| | Longformer | 512 | ✓ | 84.1 | 83.2 |
| | | | - | **98.2** | 98.1 |
| | | 1024 | ✓ | 85.0 | 84.5 |
| | | | - | **98.2** | **98.2** |
| HND | BERT | 512 | ✓ | 52.2 | 34.5 |
| | | | - | 72.6 | 70.6 |
| | Longformer | 512 | ✓ | 56.6 | 36.1 |
| | | | - | **79.0** | **78.1** |
| | | 1024 | ✓ | 56.6 | 36.1 |
| | | | - | 77.2 | 75.5 |

Table 11: **Maximum input length effect on performance.** Accuracy and macro averaged F1-score for BERT and Longformer variants are reported as an average over 10 independent executions. A checkmark in the FZ column indicates that the corresponding results were obtained by freezing the model.

| | | Execution Time [$s$] | | |
|---|---|---|---|---|
| **Strategy** | **Dataset** | **Gen** | **Run** | **Total** |
| Intuitive Graph | App Reviews | 29.1 | 139.7 | 168.8 |
| | DBpedia | 92.7 | 291.6 | 384.3 |
| | IMDB | 256.5 | 217.1 | 473.6 |
| | BBC News | 101.6 | 69.1 | 170.6 |
| | HND | 48.8 | 67.4 | 116.3 |
| TextLevel GCN | App Reviews | 71.1 | 475.3 | 546.4 |
| | DBpedia | 93.7 | 333.2 | 426.8 |
| | IMDB | 487.4 | 534.8 | 1,022.3 |
| | BBC News | 271.2 | 413.1 | 684.2 |
| | HND | 193.5 | 233.3 | 426.8 |

Table 12: **Graph execution time.** Average execution time for Intuitive Graph and TextLevelGCN approaches. It includes graph generation time (Gen) and GNN running time (Run).

| Method | App Reviews | | DBpedia | | IMDB | | BBC News | | HND | |
|---|---|---|---|---|---|---|---|---|---|---|
| | Util. | Mem. | Util. | Mem. | Util. | Mem. | Util. | Mem. | Util. | Mem. |
| Window | 3.13 | 4.74 | 3.67 | 4.75 | 4.53 | 4.79 | 3.67 | 4.81 | 1.93 | 4.83 |
| Window$_{ext}$ | 3.07 | 4.74 | 3.60 | 4.75 | 4.73 | 4.83 | 4.33 | 4.84 | 2.53 | 4.89 |
| Sequence | 2.87 | 4.74 | 2.87 | 4.74 | 3.93 | 4.79 | 3.67 | 4.79 | 2.47 | 4.83 |
| Sequence$_{simp}$ | 3.07 | 4.74 | 3.73 | 4.74 | 4.27 | 4.79 | 3.67 | 4.79 | 2.00 | 4.82 |
| TextLevelGCN 1-g | 4.07 | 5.00 | 6.73 | 5.12 | 12.33 | 5.56 | 9.07 | 5.41 | 6.13 | 5.21 |
| TextLevelGCN 2-g | 3.93 | 5.00 | 6.80 | 5.13 | 13.40 | 5.56 | 8.20 | 5.55 | 4.60 | 5.29 |
| TextLevelGCN 3-g | 3.67 | 5.00 | 6.53 | 5.13 | 10.40 | 5.71 | 6.80 | 5.58 | 3.93 | 5.37 |
| TextLevelGCN 4-g | 4.33 | 5.00 | 5.40 | 5.13 | 9.53 | 5.86 | 4.13 | 5.62 | 3.93 | 5.32 |
| BERT | 94.47 | 29.42 | 94.70 | 29.42 | 95.27 | 29.42 | 89.40 | 29.42 | 68.93 | 29.42 |
| Longformer | 99.27 | 67.86 | 99.27 | 67.86 | 99.60 | 67.86 | 99.80 | 67.86 | 99.40 | 67.86 |

Table 13: **GPU statistics (%).** GPU utilization (**Util.**) and GPU memory usage (**Mem.**) for each of the studied models. The $i$-g notation accompanying TextLevelGCN stands for $i$-gram graph construction.

| Library | Version |
|---|---|
| datasets | 2.4.0 |
| gensim | 4.2.0 |
| nltk | 3.7 |
| numpy | 1.23.1 |
| pytorch-lightning | 1.7.4 |
| scikit-learn | 1.1.2 |
| torch | 1.11.0 |
| torch-cluster | 1.6.0 |
| torch-geometric | 2.1.0 |
| torch-scatter | 2.0.9 |
| torch-sparse | 0.6.15 |
| torch-spline-conv | 1.2.1 |
| torchmetrics | 0.9.3 |
| torchvision | 0.12.0 |
| transformers | 4.21.2 |
| word2vec | 0.11.1 |

Table 14: **Libraries.** Versions of Python libraries used for the experimental implementation.