# OpenReview forum: "Connecting the Dots: What Graph-Based Text Representations Work Best for Text Classification using Graph Neural Networks?"
_EMNLP/2023/Conference — EMNLP 2023 Findings_

### Official Review · Reviewer_LsjL · 2023-08-02

**Soundness:** 3

**Excitement:**

4: Strong: This paper deepens the understanding of some phenomenon or lowers the barriers to an existing research direction.

**Paper Topic And Main Contributions:**

The authors investigate the usage of graph representation methods for text classification and empirically show that graph methods are efficient and effective for solving text classification tasks, highlighting their potential as a valuable alternative to classical machine learning and graph mining approaches in modern settings. They perform an extensive analysis by using 3 types of convolutional layers as different message-passing strategies for 12 GNN architectures. Some of these GNNs also use pre-trained word embeddings as node features. The authors also present  2 transformer-based large language models for complementing the study and providing a comparison with traditional GNN approaches. The authors note that simple graph constructions perform better as document length increases, demonstrating that graph representations are particularly beneficial for longer documents compared to Transformer-based models.



**Questions For The Authors:**

- The authors do not mention if they include the graph construction step to calculate the execution time.

**Reasons To Accept:**

- Comprehensive analysis: The authors perform an extensive analysis by considering three types of convolutional layers as different message-passing strategies for 12 Graph Neural Network (GNN) architectures.

- GNN applicability in non-graph datasets is a growing body of research and this work furnishes multiple baselines highlighting the potential of these models as effective alternatives to traditional methods.



**Reasons To Reject:**

- Graph based methods for text classification introduce an extra pre-processing step to convert sequential data to graphs. I don't see a lot of performance gains given the overhead (except in long pieces of text)

- General performance (table 4) indicates that graph models initialized with BERT embeddings almost always perform worse as compared to BERT, What is the intuition behind this?

- The authors don't provide a clear reasoning as to why structural information could be useful for text classification of datasets mentioned. For instance, how would the graph structure capture sentiment (IMDB) in a better manner as compared to traditional NLP approaches mentioned



**Reproducibility:**

4: Could mostly reproduce the results, but there may be some variation because of sample variance or minor variations in their interpretation of the protocol or method.

**Reviewer Confidence:**

4: Quite sure. I tried to check the important points carefully. It's unlikely, though conceivable, that I missed something that should affect my ratings.

---

> ### Author Rebuttal · Authors · 2023-08-28
>
> We would like to express our sincere gratitude for your thoughtful assessment and feedback, which is very valuable for improving our work. We address all the points detailed in your review as follows.
>
> #### **Performance Gains**
>
> Thanks for sharing your concern with us. We have already provided details regarding the performance of the corresponding models in Table 4. We recently also measured the GPU utilization of the studied models. There are two important points to be made:
> 1. Despite a performance gap between graph strategies and LLMs when dealing with short texts as in App Reviews, this gap becomes narrower as documents become longer. Graph representations aim to exploit the structure of a given text, so they are expected to perform better on longer documents, since there is more information and structural patterns to exploit.
> 2. Even when an additional step is needed to create the graph representation of a document, our experiments indicate that the total execution time (graph creation step + model running) is not a concern. For example, for short texts as in DBpedia, the window graph is competitive with the best performing LLM, showing a difference of only 0.8% of accuracy and being 5.7 times faster. Similarly, BERT beats Sequence graph in IMDB by only 0.5% in accuracy, while being 3.5 times slower.
>
> We will be glad to extend this analysis in our paper to showcase the behavior of graphs.
>
> #### **BERT initialization**
> Since both models use pre-trained BERT embeddings as initialization, we can note that the differences are due to how both models operate.
>
> The GNN models considered in our study are simple in terms of architecture and number of hyperparameters when compared to BERT. Therefore, they tend to fail when aggregating information from neighborhoods described by higher-dimensional vectors (BERT/BERT-C with 768 dim.). Instead, simpler vectors such as Word2Vec or Glove (300 dim.) allow GNNs to better share and extract information between different neighborhoods via message passing, being able to exploit the local context as well as the global context of the represented documents.
>
> In contrast, BERT can process more specialized vectors, as it has a much larger number of parameters, more neural layers, and a quadratic attention module. Having a quadratic attention module is comparable to having a fully connected graph model, being able to attend to every single pair of elements of the input text.
>
> #### **Why is structural information useful?**
> Document comprehension involves interpreting words that can alter the meaning of the text based on their placement. For example, in "*The movie was boring, but I was surprised by the ending*", the word "*but*" contrasts ideas. Graphs can capture this complexity, representing associations and dependencies among text units, leveraging language structure. Unlike traditional models, GNNs are capable of capturing information over longer distances through message passing, in considerably fewer steps, as Figure 2 shows.
>
> It is important to emphasize that, although graph models are competitive for solving text classification tasks in different domains, they are especially beneficial for sentiment analysis and hyperpartisan detection, benefitting from local context exploration.
>
> We will extend the Introduction to clarify the motivation for using graph representations for text classification. Thank you for bringing this to our attention.
>
> #### **Question about Execution Time**
> The graph construction step is included in our execution time calculation. Graph creation and model runtime details are provided in Appendix D, Table 12.
>
> In Section 4.3, we suggested referring to Appendix D for runtime details, but we will also include this information in Table 4 to enhance clarity. We appreciate your observation.

---

### Official Review · Reviewer_KR2u · 2023-08-04

**Soundness:** 3

**Excitement:**

3: Ambivalent: It has merits (e.g., it reports state-of-the-art results, the idea is nice), but there are key weaknesses (e.g., it describes incremental work), and it can significantly benefit from another round of revision. However, I won't object to accepting it if my co-reviewers champion it.

**Missing References:**

The references cited in this paper are not up-to-date, with only two references from the year 2022 being the most recent. If possible, the authors are advised to conduct further research on more recent studies related to the relevant task, especially the latest research concerning text classification.

**Paper Topic And Main Contributions:**

The paper highlights the exploration of Graph Neural Networks (GNNs) in the context of text classification, considering their demonstrated success in machine learning. The researchers extensively investigated various graph representation methods and text representation methods, revealing practical implications and identifying open challenges in this area. The study compares different text classification schemes across multiple datasets, considering varying document lengths. Additionally, the inclusion of two Transformer-based large language models complements the analysis.

**Questions For The Authors:**

1. This paper primarily demonstrates the effectiveness of Graphs in text classification tasks, an aspect that has been discussed in various previous works.

2. The experimental results in this paper are primarily based on the existing work and reproduction of TextLevelGCN. However, it remains unclear what specific contributions the authors have made in this domain. If there are any, further clarification is needed.

3. Why did the author only use the basic 'bert-uncased-base' model in the experiments and did not explore other potentially superior pre-trained models for performance validation?

4. The authors did not present performance evaluations when using graphs to model different text datasets. There is a concern about the potential increase in machine resource utilization when employing graphs to represent textual data, which could hinder practical usage in real-world scenarios. The authors are encouraged to provide further explanations on this matter.

Please provide an explanation for the concerns mentioned above.

**Reasons To Accept:**

The results indicate several key findings: Firstly, the effectiveness of graphs depends on the textual input features and domain, with simple graph constructions performing better as the documents become longer. Secondly, graph representations show particular benefits for longer documents, outperforming Transformer-based models. Thirdly, graph methods demonstrate notable efficiency in addressing the text classification task.

**Reasons To Reject:**

The author may need to further consider potential limitations, such as the specific selection of datasets and the need for further investigation into the generalizability of the findings across different text classification tasks and domains.

Please refer to the questions for the authors.

**Reproducibility:**

3: Could reproduce the results with some difficulty. The settings of parameters are underspecified or subjectively determined; the training/evaluation data are not widely available.

**Reviewer Confidence:**

4: Quite sure. I tried to check the important points carefully. It's unlikely, though conceivable, that I missed something that should affect my ratings.

---

> ### Author Rebuttal · Authors · 2023-08-29
>
> We sincerely appreciate the time you dedicated to reviewing our paper. Your comments are warmly welcomed. In the following lines, we address each of your suggestions and concerns.
>
> #### **Generalizability of the findings**
> We agree that the investigation ought to be conducted across a wide range of scenarios to ensure the generalizability of results. Since most of the previous work only focused on topic classification on short documents (Reuters 21578, Ohsumed, 20Newsgroups), our study specifically had the goal of assessing the graph strategies on a wider range of text classification tasks: sentiment analysis, topic classification, and hyperpartisan detection in balanced and unbalanced scenarios (including much longer documents as well).
>
> Since our results show that graph effectiveness is highly dependent on textual features and domain, we are enthusiastic about extending our study to further tasks in future work. To this end, we are releasing our code on GitHub and hope that it can grow to become a community resource covering even more tasks.
>
> #### **Question regarding previous work**
> Indeed, numerous prior studies have demonstrated graphs' efficacy in the field. However, they only validated the effectiveness on a small range of tasks with only short documents. Instead, we seek to comprehensively assess the merits and drawbacks of different previously proposed document graphs and variants in more diverse scenarios.
>
> Text classification extends beyond topic classification, encompassing real-world challenges like noisy texts, imbalanced labels, and much longer texts consisting of more than a few paragraphs. Our experiments for evaluating how graphs generalize to other tasks and our findings substantiate our contributions to the field.
>
> #### **Contributions**
> We understand the reviewer's concern and are happy to outline our contributions. We will extend the corresponding sections to clarify the paper's arguments and findings.
> 1. We evaluate previously proposed graph construction methods for text classification on a broader range of tasks, domains, and datasets.
> 2. We evaluate the efficacy vs. efficiency of the proposals, an aspect usually neglected in previous studies. We found that graphs can hardly outperform Transformer-based LLMs, but are particularly efficient in terms of execution time as well as resource utilization.
> 3. We found that document graphs are especially beneficial for long-text classification, an aspect unexplored in prior research.
> 4. We identify limitations regarding the graph construction, a nuanced process dependent on text features and data domain. Moreover, based on our experimental results, we provided a discussion regarding what graph construction and GNN architecture choice is preferable depending on the task to be solved.
>
> #### **Pre-trained models**
> As mentioned above, the goal of our study is not to obtain state-of-the-art results on the respective datasets using the latest LLMs, but rather to consider the merits of graphs, which also require far fewer computational resources.  Hence, results with BERT-base are provided merely as an informative point of reference for comparison.  We further included Longformer-base as a pre-trained language model that can cope with longer sequences of text as well. This aspect was desirable when comparing LLMs to graphs, which use the complete source text.
>
> ####  **Performance Evaluation**
> We have measured the GPU utilization and GPU memory usage for each of the models presented in Table 4 to complement the already reported performance metrics. We also measured these metrics for each graph construction when applied to each of the datasets to find out how the strategies behave when scaling to longer documents (*we rerun all the models using the same batch size for a fair comparison*). The results suggest:
> * The increase in GPU utilization is minimal as the document length increases. Specifically, as the document length increases by one order of magnitude, GPU utilization increases by about 1.5% when employing Intuitive graphs and 8-10% for TLGCN.
> * The GPU usage of graph strategies is limited to only 5% (Intuitive graphs) and 14% (TLGCN) of that used by Transformer-based language models, an important aspect when computational resources are limited.
>
> We would be pleased to integrate the analysis of our newly obtained results into the discussion of runtime in Section 4.3.
>
> #### **References**
> We will include three recent articles in our literature review section.
>
> [1] is a heterogeneous graph construction, introducing topic nodes to enhance class-aware representation learning. However, it has the same limitations as TextGCN (high computing resource consumption and transductive learning). Hence, learning for new documents is not possible without re-training the model).
> Alternatively, [2,3] are inductive models that have reported good results on traditional text classification benchmarks, but the improvement is mostly due to the combination of GNN and BERT models. Thus, these strategies are resource-intensive, hard to apply to long documents, and beyond the scope of our study.
>
> - [1] Gu, Y., Wang, Y., Zhang, H. R., Wu, J., & Gu, X. (2023). Enhancing Text Classification by Graph Neural Networks With Multi-Granular Topic-Aware Graph. IEEE Access, 11, 20169-20183.
> - [2] Huang, Y. H., Chen, Y. H., & Chen, Y. S. (2022, October). ConTextING: Granting Document-Wise Contextual Embeddings to Graph Neural Networks for Inductive Text Classification. In Proceedings of the 29th International Conference on Computational Linguistics (pp. 1163-1168).
> - [3] Wang, Y., Wang, C., Zhan, J., Ma, W., & Jiang, Y. (2023). Text FCG: Fusing contextual information via graph learning for text classification. Expert Systems with Applications, 119658.
>
> #### **Reproducibility**
> To ensure the reproducibility of our research, we have provided supplementary experiment details in Appendix A, C, and E, regarding the Python framework, further model hyperparameters, and GPU details. If preferable, we can move some of this information to Section 3.3 to support the reader's understanding.

---

### Official Review · Reviewer_bG3X · 2023-08-05

**Soundness:** 3

**Excitement:**

4: Strong: This paper deepens the understanding of some phenomenon or lowers the barriers to an existing research direction.

**Paper Topic And Main Contributions:**

This article mainly explores the advantages and limitations of using graph representation learning in text classification. The article compares different graph construction methods and various graph-based text representation techniques, evaluating their performance in text classification tasks. Additionally, it compares these methods with mainstream non-graph-based text classification approaches to investigate the applicability and advantages of graph representation learning in text classification tasks.

**Questions For The Authors:**

Section 4.1 mentions GCN and GIN and says the properties of each, that GIN will be better than GCN, how should this be interpreted in a text classification task and why is GIN better than GCN in text categorization?

**Reasons To Accept:**

1.	This article systematically explores the performance of using Graph Neural Networks (GNN) for graph representation learning in text classification. Furthermore, it compares GNN with other mainstream text classification methods, summarizing the advantages and limitations of GNN in the field of text classification.
2.	This article provides a comprehensive experimental design and a particular explanation of the experiments conducted.


**Reasons To Reject:**

1.	The introduction of this article lacks a clear explanation of its motivation, and it is essential to provide necessary references to previous related works when mentioned.
2.	In the second section, while many related works are presented, the lack of logical coherence results in a somewhat disorganized content.
3.	Regarding Section 4.1, I believe the author did not sufficiently explain how the architecture and settings of GNN impact the results of text classification. In this section, the author only presented the experimental results and the GNN used, without effectively integrating the text classification task with the chosen GNN characteristics for a comprehensive explanation.


**Reproducibility:**

3: Could reproduce the results with some difficulty. The settings of parameters are underspecified or subjectively determined; the training/evaluation data are not widely available.

**Reviewer Confidence:**

4: Quite sure. I tried to check the important points carefully. It's unlikely, though conceivable, that I missed something that should affect my ratings.

---

> ### Author Rebuttal · Authors · 2023-08-28
>
> We are thankful for your valuable suggestions and truly appreciate the thorough review. We address your comments and concerns as follows.
>
> #### **Introduction**
> We will revise the relevant paragraph to enhance the clarity of our motivation. Specifically, there are  three key points we will highlight, while substantiating our claims with the appropriate references:
> 1. Most previously proposed approaches were created with a focus on specific tasks within specific domains.
> 2. The validation of previous graph proposals was performed on a limited range of datasets and model architectures.
> 3. Earlier graph proposals were developed before Graph Neural Networks (GNNs) emerged and validated using graph mining techniques.
>
> #### **Related Work**
> To present related work in a more logically structured manner, we will categorize the information into three subsections based on the type of graph representation proposed, as follows:
> * Early Graph Constructions: The first proposed graph constructions, based mainly on co-occurrence and statistics, and validated using graph mining.
> * Linguistic Features as Graphs: Constructions integrating lexical, semantic, or syntax information as more advanced graph representations.
> * Specialized Graph Constructions: Recently proposed methods, including more complex constructions and integrating Graph Convolutional Neural Networks (GCNs) in the learning process.
>
> #### **How GNN architecture affects the results**
> We agree that a finer analysis regarding the choice of the GNN architecture is essential to fully comprehend a model’s effectiveness under diverse conditions. Due to space constraints, we included experiments with different numbers of layers in Appendix B, focusing our discussion on message-passing methods and node initialization.
>
> Our results suggest that effective node feature initialization yields good results with fewer hidden dimensions in most tasks. Exceptions are observed on App Reviews and HND due to their intricacies, as outlined in Section 4.2.
> Additionally, the number of convolutional layers is linked to the task. If the task focuses on local patterns, using 1 or 2 layers is recommended. But if the task needs broader information, more layers will be needed. Yet, understanding context (local/global) varies with document length. For instance, to comprehensively grasp the document's sentiment, a sentence-level analysis is vital, whereas if the document comprises only one or two sentences, a wider view will be required. This is shown in Table 2 and Table 5, where using 3 layers produced the best App Reviews results.
>
> #### **Question about GIN**
> As stated in Section 4.1, given a node and its neighbors, GCN computes the average of all those representations in one undifferentiated step. Conversely, GIN's weighted summation aggregates neighbor node representations to the target node while incorporating a learned variation (epsilon). As a result, GIN distinguishes node neighborhoods, discerning structural information among graph classes, unlike GCN.
> Since our document graphs are based on word co-occurrence, this means that GIN can exploit structural patterns of the co-occurrences and thus identify recurrent associations between specific words, which can be crucial to predict the correct label of the graph.
>
> #### **Reproducibility**
> To ensure the reproducibility of our research, we provided supplementary experimental details in Appendix A, C, and E, regarding the Python framework, further model hyperparameters, and GPU details. If preferable, we can move some of this information to Section 3.3 to support the reader's understanding. Please also note that we will release our code in a publicly available GitHub repository once the Anonymity Period ends.

---

### Meta-Review · Area_Chair_wQ95 · 2023-09-19

**Recommendation:** 3

**Metareview:**

The paper investigates the use of Graph Neural Networks (GNNs) in text classification and conducts a thorough analysis of different graph representation and text representation methods. The study compares the performance of GNNs with mainstream text classification methods and addresses questions related to their efficiency and effectiveness for text classification tasks. The paper's strengths lie in its comprehensive experimental design, extensive analysis, and clear presentation. However, there are some concerns about the motivation, the original contributions, and the lack of explanation regarding the choice of pre-trained models.

One of the paper's significant strengths is its comprehensive analysis of GNNs for text classification, covering various GNN architectures and message-passing strategies. This thorough examination provides valuable insights into the applicability of GNNs in this domain. Additionally, the study's systematic comparison with other text classification methods enhances its overall quality.

However, the paper faces some challenges. First, the motivation for the research and its original contributions should be clarified further. The paper appears to build upon existing works without explicitly stating its unique contributions to the field. Second, the choice of using the 'bert-uncased-base' model in experiments without exploring potentially superior pre-trained models needs justification. Third, while the study demonstrates the efficiency and effectiveness of GNNs for text classification, there is a need for more extensive performance evaluations on different text datasets to assess potential resource utilization issues. Lastly, the paper should provide a clearer rationale for why structural information captured by GNNs is beneficial for specific text classification tasks, such as sentiment analysis.

---

### Decision · Program_Chairs · 2023-10-07

**Decision:**

Accept-Findings

**Comment:**

The paper investigates the use of Graph Neural Networks (GNNs) in text classification and conducts a thorough analysis of different graph representation and text representation methods. The study compares the performance of GNNs with mainstream text classification methods and addresses questions related to their efficiency and effectiveness for text classification tasks. The paper's strengths lie in its comprehensive experimental design, extensive analysis, and clear presentation. However, there are some concerns about the motivation, the original contributions, and the lack of explanation regarding the choice of pre-trained models.

One of the paper's significant strengths is its comprehensive analysis of GNNs for text classification, covering various GNN architectures and message-passing strategies. This thorough examination provides valuable insights into the applicability of GNNs in this domain. Additionally, the study's systematic comparison with other text classification methods enhances its overall quality.

However, the paper faces some challenges. First, the motivation for the research and its original contributions should be clarified further. The paper appears to build upon existing works without explicitly stating its unique contributions to the field. Second, the choice of using the 'bert-uncased-base' model in experiments without exploring potentially superior pre-trained models needs justification. Third, while the study demonstrates the efficiency and effectiveness of GNNs for text classification, there is a need for more extensive performance evaluations on different text datasets to assess potential resource utilization issues. Lastly, the paper should provide a clearer rationale for why structural information captured by GNNs is beneficial for specific text classification tasks, such as sentiment analysis.